# Transcriptome and Biochemical Analyses of a Chlorophyll-Deficient Bud Mutant of Tea Plant (*Camellia sinensis*)

**DOI:** 10.3390/ijms242015070

**Published:** 2023-10-11

**Authors:** Yuanyuan Li, Chenyu Zhang, Chunlei Ma, Liang Chen, Mingzhe Yao

**Affiliations:** Key Laboratory of Biology, Genetics and Breeding of Special Economic Animals and Plants, Ministry of Agriculture and Rural Affairs, Tea Research Institute of the Chinese Academy of Agricultural Sciences, Hangzhou 310008, China; liyuanyuan@tricaas.com (Y.L.); zhangchenyu@tricaas.com (C.Z.); malei220@tricaas.com (C.M.); liangchen@tricaas.com (L.C.)

**Keywords:** amino acid, bud mutation, *Camellia sinensis*, chlorophyll biosynthesis, flavonoid, RNA-seq

## Abstract

Tea leaf-color mutants have attracted increasing attention due to their accumulation of quality-related biochemical components. However, there is limited understanding of the molecular mechanisms behind leaf-color bud mutation in tea plants. In this study, a chlorina tea shoot (HY) and a green tea shoot (LY) from the same tea plant were investigated using transcriptome and biochemical analyses. The results showed that the chlorophyll a, chlorophyll b, and total chlorophyll contents in the HY were significantly lower than the LY’s, which might have been caused by the activation of several genes related to chlorophyll degradation, such as *SGR* and *CLH*. The down-regulation of the *CHS*, *DFR*, and *ANS* involved in flavonoid biosynthesis might result in the reduction in catechins, and the up-regulated *GDHA* and *GS2* might bring about the accumulation of glutamate in HY. RT-qPCR assays of nine DEGs confirmed the RNA-seq results. Collectively, these findings provide insights into the molecular mechanism of the chlorophyll deficient-induced metabolic change in tea plants.

## 1. Introduction

Tea (*Camellia sinensis* (L.) O. Kuntze) is a perennial woody plant whose young and fresh leaves are primarily utilized for producing the homonymous beverage. The color of a tea’s fresh leaves not only determines the visual attributes of the final tea product but also correlates with its physicochemical characteristics impacting the flavor of the processed tea, thereby exerting a considerable influence on the economic benefits [1]. Hence, the chlorophyll-deficient mutants of tea plants have gradually attracted increasing attention due to their distinct historical background, specific visual characters related to the color of the processed tea, and extraordinary tea flavor with increasing umami taste and decreasing astringency, thereby exerting considerable influences on their economic benefits [2,3].

According to the degree of chlorosis phenotype, the chlorophyll-deficient mutants of tea plants can be classified into three collections: albino, chlorina, and variegation, and can also be subcategorized with respect to their regreening mechanisms into light- and temperature-sensitive types [1]. According to a previous study, an altered yellow leaf color is mainly caused by the reduction in chlorophyll content and the abnormal development of chloroplasts in irregular ultrastructure [4,5,6]. Researchers have found that repressed genes encoding photosynthetic antenna proteins such as Lhca and Lhcb were closely associated with the aberrant chloroplast development in the yellow leaf variation of tea plants [7]. In ‘Menghai Huangye’, which is a type of albino tea in Yunnan, the down-regulated gene expression of *HEME2* and *POR*, related to chlorophyll biosynthesis, might block the biosynthesis of the chlorophyll precursors, thus making it highly connected to the low chlorophyll content of this variety of tea [8]. However, the concrete molecular mechanism of the chlorophyll-deficient phenotype needs further research.

Furthermore, the leaf color changes of these specific chlorina tea plant germplasms are normally accompanied by changes in the internal biochemical components, presenting higher concentrations of total amino acids, especially theanine, but lower catechins [9,10,11] compared to the traditional green-leaves tea germplasms. These different quality-related components are closely associated with the attractive tea qualities of brisk flavor and reduced astringency and bitterness in the processed tea from these chlorotic tea plants [12]. Some studies have shown the potential molecular mechanisms of these biochemical changes. Transcriptome and correlation analyses have indicated that *CsGS1*, *CsPDX2*, *CsGGP5*, *CsHEMA3*, and *CsCLH4* are potential key genes that may be responsible for the differential accumulation of theanine in green and yellow tea cultivars [13]. Moreover, the critical interval of candidate genes, including GDH, related to amino acid metabolites was mined through quantitative trait loci analysis [14]. A recent study showed that the accumulation of theanine in albino or etiolated tea cultivars might due to the strengthened biosynthesis and weakened catabolism of theanine, which were probably caused by the activated expression of *CsAlaDC* and *CsGOGAT1*, as well as by the decreased expression of *CsGGT2* [15]. In chlorotic “Huangjinya” tea leaves, the down-regulated expression of *PAL* and *4CL* (encoding the rate-limiting enzymes of the flavonoid biosynthesis pathway) in both gene-expression and protein-expression levels might be the reason for the inhibition of flavonoid accumulation [16]. Although there were several studies that suggested the relationship between the molecular and biochemical mechanism, the specific causes of the biochemical changes remain unclear.

In this paper, studies of a chlorina shoot (HY) and green tea shoot (LY) of ‘Danzicha’, with the two materials sharing almost the same genetic background, were performed using transcriptome and biochemical analyses to illustrate the mechanism of the etiolated phenotype’s and different metabolites’ formation. The DEGs involved in the leaf-color mutation were identified using transcriptome analysis, and the different biochemical components were measured using biochemical analysis. This research provides an understanding of the analysis and application of bud mutations in tea plants.

## 2. Results

### 2.1. Morphological Features and Pigment Contents’ Comparison in HY and LY

As shown in Figure 1A, the leaf color of LY was green, whereas its bud mutant shoot HY exhibited chlorina leaves during spring. Further pigment measurements showed a significant reduction in the levels of total chlorophylls (47.09% lower than that in LY), Chl a (28.86% lower), Chl b (79.94% lower), and the Chl/Car ratio (71.26% lower) in HY compared to LY. However, the concentration of total carotenoids (84.15% higher than that in LY) and the ratio of Chl a/b (255.36% higher) in HY exhibited significantly higher levels than those in LY (Figure 1B).

It was observed that the reduction in chlorophyll content and the enhancement of total carotenoid levels aligned with the observed phenotypic alterations in the chlorina leaves, indicating that the chlorophyll-deficient bud mutant might have been caused by the alteration of chlorophylls and carotenoid content.

### 2.2. Transcriptome Profiling of HY and LY

#### 2.2.1. Evaluation of the Transcriptome Quality

HY and LY are the materials on the same tea plant with almost the same internal genetic background and an external growth environment including the same nutritional conditions; therefore, the transcriptome profiling in both HY and LY were compared for a better understanding of the differences between the yellow bud mutation and the original green shoots at the transcription level, removing the other disturbing factors of leaf color changes such as different genetic backgrounds and environmental impacts.

A total of 263,819,288 clean reads were obtained, with 130,285,332 reads from the HY shoot and 133,533,956 reads from the LY shoot. The GC contents of different samples ranged from 43.86% to 44.85% and the Q20 percentages measured 97.26% and above. The paired-end clean reads were aligned to the reference ‘Huangdan’ genome sequences, where the mapping ratio of each sample ranged from 88.60 to 89.94% (Appendix A). Collectively, all the subsequent analyses were based on the high quality of sequencing results. High Pearson’s correlation coefficients (R2) over 0.922 were maintained between the biological replicate samples in the same group, and lower R2 were maintained between the samples in different groups (Figure 2A). A principal component analysis (PCA) showed that three biological replicates of each group clustered together and that HY clustered apart from LY in the first principal component (Figure 2B). These results indicated highly correlated gene expression profiles among biological replicates within groups and also showed obvious differences between HY and LY.

#### 2.2.2. DEGs Analysis of HY and LY

Differential expression analysis was performed based on the thresholds of |log_2_(Fold Change)| ≥ 1 and padj (a common form of false discovery rate (FDR)) ≤ 0.05. There were 2810 (1245 up- and 1565 down-regulated) DEGs in HY compared to LY. As shown in Appendix A, GO enrichment indicated that under the cellular component (CC) category, the up-regulated DEGs were enriched in the terms of photosystem II oxygen-evolving complex, thylakoid membrane, photosystem II, and that the down-regulated DEGs were significantly (padj < 0.05) enriched in the terms of magnesium–ion binding and other terms under the molecular function (MF) category. In the KEGG enrichment analysis, the down-regulated DEGs were enriched through several pathways, such as carbon fixation in photosynthetic organisms, flavonoid biosynthesis, and other pathways.

#### 2.2.3. DEGs in Photosynthesis

The GO and KEGG enrichment analyses exhibited that multiple DEGs in HY compared to LY were enriched in the terms of photosynthesis and its related pathways, such as the following: photosynthesis, the photosystem II oxygen-evolving complex, photosystem II, the photosystem and photosynthetic membrane in GO terms and the carbon fixation in photosynthetic organisms, the photosynthesis-antenna proteins, and the photosynthesis in the KEGG pathways. Further analysis of these GO terms found that two up-regulated genes (*TGY001383*, encoding the photosynthetic NDH subunit of lumenal location, and *TGY108318*, encoding the PsbP domain containing protein 6) were assigned to multiple terms related to photosynthesis and components of the photosystem in the CC and BP category. Among these KEGG pathways, fourteen up-regulated and thirteen down-regulated genes were identified, including twelve up-regulated and twelve down-regulated genes of carbon fixation in the photosynthetic organisms’ pathways, one up-regulated gene encoding the F-type H^+^-transporting ATPase subunit gamma (TASG) in the photosynthesis’ pathway, one up-regulated gene encoding light-harvesting complex II chlorophyll a/b binding protein 6 (Lhcb6), and one down-regulated gene encoding Lhcb3 in the photosynthesis-antenna proteins’ pathway. These results indicate that changes in the expression levels of these genes might have induced the changes in the photosynthesis process.

#### 2.2.4. DEGs in Pigment Biosynthesis

In the porphyrin and chlorophyll metabolism pathway (Figure 3A,C), three DEGs were identified, including one up-regulated gene encoding the STAY-GREEN protein (magnesium dechelatase) and two genes annotated as chlorophyllase (CLH) (one of them was up-regulated and the other was down-regulated) in HY compared to LY. In the carotenoid biosynthesis pathway (Figure 3B,C), there were one up-regulated gene encoding (+)-abscisic acid 8’-hydroxylase (AAH) and three down-regulated genes, containing two genes that encode xanthoxin dehydrogenase (ABA2) and one gene that encodes zeaxanthin epoxidase (ZEP). The changes in gene expression in the porphyrin and chlorophyll metabolism and in the carotenoid biosynthesis pathways suggested that th;oe degradation of the chlorophyll seems to be strengthened and the conversion of carotenoids might be reduced in the HY shoot, ultimately resulting in the phenotype characteristics of HY and LY.

#### 2.2.5. DEGs in Flavonoid and Amino Acid Biosynthesis

Among the down-regulated DEGs, flavonoid biosynthesis was the most significantly enriched subcategory in KEGG pathway enrichment. Eight down-regulated genes were identified, including two genes encoding anthocyanidin synthase (ANS), two genes that encode dihydroflavonol 4-reductase (DFR), two genes that encode caffeoyl-CoA *O*-methyltransferase (CCoAOMT), one gene that encodes shikimate *O*-hydroxycinnamoyltransferase (HCT), and one gene that encodes chalcone synthase (CHS). The reduced expression levels of these key enzymes involved in flavonoid biosynthesis may suggest the decreasing catechins contents in the yellowing cultivar HY.

To validate this hypothesis, a further flavonoid measurement was performed, and the results exhibited significantly lower contents of catechins, including (−)-epigallocatechin (EGC), (−)-epicatechin (EC), (−)-epigallocatechin gallate (EGCG), and (−)-epicatechin gallate (ECG), but significantly higher contents of purine alkaloids, including theobromine and caffeine, in HY than LY. Among them, the EGCG exhibited the highest abundance (Figure 4C).

Moreover, the DEGs involved in amino acids’ metabolism, such as the arginine (Arg) biosynthesis and alanine (Ala), and the aspartate (Asp) and glutamate (Glu) metabolism, were screened to realize the changes in the biochemical components between HY and LY. In arginine biosynthesis (Figure 5A,C), there were six up-regulated genes containing one gene encoding glutamate dehydrogenase A (GDHA), one encoding aspartate aminotransferase (ASP), one encoding glutamate–glyoxylate aminotransferase (GGT), one encoding ornithine carbamoyltransferase (OTC), and two encoding urease (URE) and three down-regulated genes containing one gene that encodes ASP, one that encodes alanine aminotransferase (AlaAT), and one that encodes OTC in arginine biosynthesis. In addition, there were six up-regulated genes containing two genes encoding adenylosuccinate synthetase (ADSS), one encoding AlaAT, one encoding L-aspartate oxidase (AO), one encoding ASP, and one gene encoding GDH and four down-regulated genes containing one gene that encodes Alanine–glyoxylate aminotransferase (AGAT), one that encodes AlaAT, one that encodes ASP, and one that encodes glutamate decarboxylase (GAD) in the alanine, aspartate, and glutamate metabolism pathway (Figure 5B,D).

As a result of the alteration in the genes involved in the amino acid metabolism pathway at the transcriptional level, it is suggested that there might be changes in the amino acid contents at the metabolic level between HY and LY. In the arginine biosynthesis pathway, the alterations of genes participating in the arginine metabolic process at the gene expression level might lead to the distinction in arginine contents between HY and LY. Additionally, the upregulation of gene expression in glutamate biosynthesis and the downregulation in glutamate conversion are likely to increase the glutamate content. Furthermore, it can be seen from the arginine biosynthesis pathway that the alterations in glutamate content would also impact the fluctuations in arginine levels.

Further determination of amino acids showed that the detected compounds exhibited diverse levels of abundance. Among them, Thea exhibited the highest content, followed by Glu, Asp, and glutamine (Gln) as the next most abundant components. Additionally, the abundance of nine free amino acids, excluding histidine (His), was found to be higher in HY compared to LY (Figure 5E).

### 2.3. Quantitative Real-Time PCR (RT-qPCR) Analysis of Candidate Genes

To further confirm the reliability of the RNA-seq results and the expression profiles of key genes related to the mentioned metabolic pathways, nine genes were selected for analysis using RT-qPCR. The RT-qPCR results (Figure 6) exhibited similar trends to the RNA-seq data, indicating the reliability of the RNA-seq results and demonstrating changes in the expression levels of key genes in important metabolic pathways related to leaf color and internal biochemical components.

## 3. Discussion

Recently, the leaf color variation of tea plants with chlorina or albino leaves has attracted increasing attention due to their distinct leaf color, as well as unique flavor and aroma. However, the molecular mechanism behind the formation of their chlorina phenotype remains unclear. Previous studies on leaf color variation and quality-related biochemical components have primarily utilized materials with different genetic backgrounds. HY, which was a spontaneous chlorina bud sport shoot of the normal green-leaf tea cultivar ‘Danzicha’, shares almost the same genetic background with the green shoot of ‘Danzicha’, making it the ideal research object of the etiolated mechanism and for the valuable breeding of germplasm resources. Moreover, this unique bud sport has been propagated by cutting into a novel cultivar of ‘Zhongcha 133’ (accession number: CNA20140553.1) and proven using a severe DUS test (D: distinctness, inter-cultivar variation; U: uniformity, intra-cultivar homogeneity; and S: stability, homogeneity of generations), where its chlorina leaf color variation can be stably maintained and has a specific commercial value. Remarkably, the transcriptome analysis showed that there was a high Pearson’s coefficient between the ‘Zhongcha 133′ and HY samples, and the two groups clustered together in the PCA analysis in the first principal component, suggesting that ‘Zhongcha 133′ and HY have a similar transcriptome composition. Meanwhile, the minimal number of DEGs in HY vs. ZC133 and the cluster analysis also provided further evidence of the similar transcriptome composition between ‘Zhongcha 133′ and HY (Appendix A). HY and LY are the materials from the same tea tree that have almost the same internal genetic background and external growth environment; therefore, only HY and LY comparisons were made for better understanding the differences between HY and LY at the transcription level. In this study, transcriptome and biochemical analyses were performed to elucidate the differences between HY (the yellow shoot of ‘Danzicha’) and LY (the green shoot) through the transcriptome profiling and their biochemical components.

### 3.1. The Underlying Mechanism of the Chlorina Phenotype in HY

Up until now, because the variation in leaf color can be readily distinguished visually, many leaf-color mutations have been found in various plants, such as cucumber [17], wheat [18], and maize [19]. These leaf-color mutations serve as the ideal research materials for the exploration of photosynthetic pigment’s biosynthesis, photosynthesis, the development of chloroplasts, and the advancement of selective breeding techniques to enhance photosynthetic efficiency [20,21]. The leaf-color mutations are primarily caused by genetic changes that affect the genes involved in the biosynthesis and degradation of photosynthetic pigments, as well as the photosynthesis pathway, chloroplast development, and differentiation. These mutated genes can directly or indirectly disrupt pigments’ synthesis and stability, leading to reduced levels of photosynthetic pigments like chlorophylls and carotenoids [21]. In the current measurement of pigments, there was a significant reduction in the levels of total chlorophylls, Chl a, Chl b, and Chl/Car, but significantly higher level of Chl a/b and total carotenoids in HY compared to LY.

Any abnormalities in the steps of the chlorophyll degradation pathway can lead to variations in the levels of chlorophyll, ultimately resulting in leaf-color mutations. Chlorophyllase is the first enzyme involved in chlorophyll degradation, which catalyzes the hydrolysis of chlorophyll into chlorophyllide and phytol [22]. There is already much research finding the relation between chlorophyllase (CLH) and leaf color mutations. The leaf-color mutants of *Cymbidium sinense* ‘Dharma’ exhibited higher expression levels of two chlorophyll degradation enzyme-coding genes (*CLH* and *red chlorophyll catabolite reductase*, *RCCR*) compared to the wild type. This coincided with a decrease in the photosynthetic pigments and an increase in the chlorophyll degradation metabolites. These findings suggest that the mutant leaves most likely resulted from chlorophyll degradation rather than biosynthesis [23]. A similar increased expression of *CLH2* was also found to be related to the reduced chlorophyll contents in a leaf-color mutant of *Cymbidium sinense* [24]. In the yellow-leaf bud mutant cultivar ‘Wannianjin’ of *Ginkgo biloba*, dynamic transcriptome analyses suggested that both the decreased expression of the biosynthesis gene, i.e., *HEMA*, and the increased expression of the degradation gene (*CLH*) caused a low chlorophylls accumulation [25]. Meanwhile, a previous study indicated that the upregulated expression of the *CLH* and *NYC1* genes in the yellow–green leaf WY16-13 of wucai (*Brassica campestris*) germplasm might have accelerated the degradation of the chlorophylls, resulting in the yellowing of the plant’s leaves [26]. In addition, STAY-GREEN (SGR) is a kind of Mg-dechelatase that has been identified in *Arabidopsis* [27] and *Chlamydomonas* [28], catalyzing the conversion of chlorophyll a to pheophytin a during the process of chlorophyll degradation. Recently, a study showed that the overexpression of *SGRL* contributed to the acceleration of chlorophyll degradation and the reduction in chlorophyll contents under dark-induced senescence in rice, exhibiting early leaf-yellowing characteristics [29]. Under different abiotic stress treatments, *SGRL*-overexpressing plants were found to exhibit early leaf-yellowing in *Arabidopsis* [30]. All of these researches suggest the potential connection between *SGR* and leaf color variation.

In this study, the transcriptome and RT-qPCR analyses showed that the *CLH* and the *SGR* were up-regulated in HY compared to LY, which implied the enhancement of chlorophyll degradation. Moreover, to further understand whether the synthesis of chlorophyll played a part in the leaf color mutation, the DEGs involved in the chlorophyll synthesis pathway were identified, and their expression patterns in HY and LY were analyzed. The RT-qPCR results indicated that six DEGs, including *Chlorophyllide a oxygenase* (*CAO*), *NADPH-protochlorophyllide oxidoreductase* (*POR*), *Mg chelatase H subunit* (*CHLH*), *Mg chelatase I subunit* (*CHLI*), *Mg chelatase D subunit* (*CHLD*), and *Genomes uncoupled 4* (*GUN4*), were up-regulated in HY compared to LY (Appendix A). Hence, taking into account the decrease in the chlorophyll levels in HY, the observed leaf-yellowing is likely attributed to the intensified degradation of the chlorophyll, rather than a deficiency in chlorophyll synthesis. The contents of the chlorophyll synthesis precursors and the chlorophyll degradation metabolites should be measured in the future to further illustrate the hypothesis.

### 3.2. Differences in the Abundances of Quality-Related Biochemical Components between HY and LY

The color of tea leaves is closely related to the sensory quality of the processed tea, which further influences its economic value [1]. Catechins are the predominant components of flavonoids, which are important sensory, quality-related, characteristic biochemical compounds in tea plants. The major catechin components include (−)-epicatechin (EC), (−)-epicatechin gallate (ECG), (−)-epigallocatechin (EGC), (−)-epigallocatechin gallate (EGCG), (+)-catechin (C), and (+)-gallocatechin (GC) [31,32]. These compounds contribute to the bitterness, astringency, and sweet aftertaste of tea beverages [32,33]. Additionally, it has been demonstrated that free amino acids, especially theanine, have an impact on the sweet, umami taste of tea [34]. Theanine is one of the most important and abundant amino acids in tea plants, and it is also a significant contributor to the flavor of processed tea. Theanine plays a key role in reducing astringency and bitterness while enhancing the brothy and sweet taste [12,35].

Previous studies have shown that the leaf color variation of tea plants commonly presented higher abundances of free amino acids, especially theanine, but lower contents of catechins and caffeine along with albino or chlorina leaves, ultimately exhibiting the excellent sensory quality of high umami, reduced bitterness and astringency, and ornamental value [9,12]. A previous study showed that, in the albino tea ‘Haishun 2’, the albino leaves exhibited significantly fewer catechins compared to the green leaves [36]. An analysis of the biochemical components revealed that ‘Anji baicha’ showed lower contents of purine alkaloids, including caffeine and theobromine, and catechins compared to ‘Longjing 43’ [37]. The yellow mutant leaves of ‘Yinghong No. 9’ were found to have significantly higher levels of L-theanine compared to the green leaves [38]. Compared to ‘Yunkang 10’, ‘Menghai Huangye’ displayed an enrichment in theanine but lower total catechin levels [39]. In this study, the transcriptome analysis showed that numerous DEGs were enriched during the flavonoid biosynthesis and amino acids metabolism-related pathways, such as the arginine biosynthesis and the alanine, aspartate, and glutamate metabolism. The differential expression of genes involved in the metabolism of catechins and amino acids may contribute to the observed variations in the metabolic contents between HY and LY. A further analysis of the biochemical components validated the changes. Similarly to the previous research, the same tendency of lower contents of catechins and higher amino acids in HY than in LY was observed.

This abnormal phenomenon of reduced flavonoids and accumulated amino acids was observed not only in the leaf color variations of tea plants but also in the *Arabidopsis* mutants with the albino or pale green phenotype [40] and in a strawberry mutant with yellow–green leaves [41]. This suggests that significant alterations have occurred in both the carbon and nitrogen metabolisms. Previous studies have shown that, in ‘Huangjinya’, the carbon metabolism pathway was inhibited, leading to impaired photosynthesis, reduced carbohydrate and flavonoid levels, and disrupted glucose metabolism. Consequently, there was an imbalance between the carbon and nitrogen metabolism in the chlorotic leaves. Additionally, the accumulation of free amino acids may be attributed to enhanced protein degradation, decreased nitrogen consumption, and efficient nitrogen resource storage [42]. The highly efficient storage of nitrogen resources is likely to be the reason for the predominance of the Glu, Thea, and Asp in albino leaves. Research on the coordinated regulation of the carbon and nitrogen metabolism in the albino tea cultivar ‘Baiye 1’ revealed that the interruption of carbon metabolism resulted in a reduced nitrogen consumption. The carbon skeletons derived from the breakdown of the nitrogen metabolism through proteolysis served as a carbon source to sustain the basic energy metabolism in the albino leaves. Furthermore, the impaired chloroplast functionality was possibly the reason for the down-regulated gene expression levels and the decreased abundance of metabolites in the flavonoid pathway [43]. In a leaf-color mutant of the strawberry species ‘Fragaria pentaphylla’, characterized by yellow–green leaves and low chlorophyll levels, researchers have speculated that the accumulation of amino acids such as Glu, Arg, and Asn may be due to the buildup of Glu-derived precursor metabolites resulting from the impaired chlorophyll synthesis [41]. Furthermore, the impaired photosynthesis resulted in the limited availability of carbon skeletons, leading to a reduction in the major end-products of carbon metabolism, such as flavonoids.

In this study, an imbalance in the carbon and nitrogen metabolism was observed in the leaf color variation HY. The reduction in flavonoids was mainly due to the impaired photosynthesis and the down-regulated expression of the DEGs involved in the flavonoid biosynthesis pathway. Additionally, the up-regulation of the gene expression involved in glutamate biosynthesis and the downregulation of glutamate conversion were observed in HY, leading to a potential increase in glutamate content. These indicated that, in addition to the aforementioned speculation, the accumulation of free amino acids, particularly glutamate, may have been due to the increased synthesis. The relationship and regulatory mechanism between the carbon and nitrogen metabolism in HY should be further analyzed in future studies.

## 4. Materials and Methods

### 4.1. Plant Materials and Sample Preparation

HY and LY are the materials on the same tea plant called *C. sinensis*, ‘Danzicha’. The plant materials HY were collected from a spontaneous yellow bud sport shoot of ‘Danzicha’, and LY was collected from the green shoot of the same tea tree. The tree was planted at the China National Germplasm Hangzhou Tea Repository (CNGHTR) of the Tea Research Institute of the Chinese Academy of Agricultural Sciences (TRICAAS), under conventional and uniform horticultural practices.

Several fresh tea shoots (one bud with two leaves) of the above two plant materials from the same tea tree acting as three biological replicates were collected randomly and stored in liquid nitrogen for transcriptome analysis in April 2022. To further clarify the molecular mechanism and the metabolic differences of the yellow bud sport mutation, other samples (one bud with two leaves) acting as three biological replicates were collected and dried for further chemical analysis in April 2023. 

### 4.2. RNA-seq

The total RNA was extracted using an Easy Pure Plant RNA Kit (Tiangen, Beijing, China), following the manufacturer’s instructions, and the details were as follows. In brief, 0.2 g of frozen samples was ground and lysed using CLB (with β-Mercaptoethanol). And then, the total RNA was isolated through adsorption and elution using special membranes. The RNA integrity was assessed using the RNA Nano 6000 Assay Kit of the Bioanalyzer 2100 system (Agilent Technologies, Santa Clara, CA, USA). The cDNA libraries were prepared and sequenced using an Illumina Novaseq platform, and 150 bp paired-end reads were generated. Clean data (clean reads) were obtained by removing reads containing adapter, reads containing ploy-N, and low-quality reads with over 50% of low-quality (Qphred ≤ 20) bases from the raw data. At the same time, the quality scores (Q20, Q30) and the guanine-cytosine (GC) content of the clean data were calculated. All the downstream analyses were based on the remaining high-quality (eliminating the low-quality reads) clean data. The paired-end clean reads with a high quality were aligned to the reference ‘Huangdan’ genome sequences [44] using Hisat2 v2.0.5 [45].

### 4.3. Enrichment Analysis of the DEGs

The read count and fragments per kilobase of transcript per million mapped reads (FPKM) value were used to normalize the quantification of gene expression level. A differential expression analysis was performed using the DESeq2R package (v 1.20.0, Bioconductor, Boston, MA, USA) and thresholds of |log_2_(Fold Change)| ≥ 1 and padj (a common form of false discovery rate (FDR)) ≤ 0.05 were set to identify significantly differentially expressed genes (DEGs). A Gene Ontology (GO database, https://www.geneontology.org/, accessed on 22 June 2022) enrichment analysis of differentially expressed genes was implemented using the cluster Profiler R package (Bioconductor, Boston, MA, USA). The GO terms with corrected *p*-value lower than 0.05 were considered significantly enriched by differentially expressed genes. The statistical enrichment of differentially expressed genes in KEGG (KEGG database, https://www.genome.jp/kegg/, accessed on 3 July 2022) pathways was tested using the cluster Profiler R package.

### 4.4. Quantitative Real-Time PCR (RT-qPCR) Analysis

To prove the dependability of the RNA-seq results, cDNA was synthesized using the PrimeScript™ RT reagent Kit with a gDNA Eraser (Perfect Real Time) (TaKaRa, Dalian, China). An RT-qPCR was performed according to the manufacturer’s instructions for the LightCycler 480 SYBR Green I master (Roche, Mannheim, Germany). The specific primers were designed through the NCBI PrimeBLAST (https://www.ncbi.nlm.nih.gov/tools/primer-blast, accessed on 26 June 2023) and were listed in Appendix A. Β-actin was selected as the internal control for normalization. The ultimate relative gene expression levels were calculated according to the 2^−ΔΔ Ct^ methods [46].

The correlative calculation equations of the relative gene expression levels were as follows:ΔΔ Ct = (C_HY, candidate gene_ − C_HY,β-actin_) − (C_LY, candidate gene_ − C_LY, Actin_)

### 4.5. Determination of Biochemical Compositions

#### 4.5.1. Determination of Chlorophylls and Carotenoids

Chlorophylls and carotenoids were extracted with 3 mL of 80% acetone from 0.02 g of dried pulverized samples, and were incubated in darkness for 24 h. The absorbance of 1 mL of extracting solution was measured at 663.2 nm, 646.8 nm, and 470 nm using a UV-2550 spectrophotometer (Shimadzu Corp., Tokyo, Japan) and was calculated referring to the methods and formula of Lichtenthaler [47]. The final contents were represented as mg/g dry weight (DW).

The content calculation equations of chlorophyll a, chlorophyll b, and carotenoids were as follows:*Chlorophyll a(Ca)* = [(12.25 × *A*_663.2_ − 2.79 × *A*_646.8_) × 3]/(*W* × 1000)
*Chlorophyll b(C_b_)* = [(21.50 × *A*_646.8_ − 5.10 × *A*_663.2_) × 3]/(*W* × 1000)
*Carotenoids* = [(1000 × *A*_470_ − 1.82 × *C_a_* − 85.02 × *C_b_*) × 3]/(*W* × 198 × 1000)
where *A*_663.2_, *A*_646.8_, and *A*_470_ represent the absorbance at 663.2, 646.8, and 470 nm, respectively, and *W* represents the DW of the samples (g).

#### 4.5.2. Determination of Free Amino Acids

Free amino acids were extracted by adding 5 mL of boiling water to 0.1 g of dried pulverized samples. The extraction was carried out in a 100 °C water bath for 30 min. The supernatant was collected through centrifuging at 3500 rpm for 10 min and then filtered through a hydrophilic nylon membrane filter with a 0.22 μm pore size. The derivatization of the free amino acids was performed using the AccQ.Tag Ultra derivatization kit (Waters, Milford, MA, USA), following the manufacturer’s instructions, and the details were as follows. In brief, the derivatization of the free amino acids was performed using 6-Aminoquinolyl-N-hydroxysuccinimidylcarbamate. The contents of the free amino acids were measured using Ultra-High Performance Liquid Chromatography (UPLC, Waters, Milford, MA, USA). The injection volume was 1 µL and the column was an AccQ·Tag Ultra Column (1.7 µm, 2.1 × 100 mm) at 43 °C. The mobile phase A used 100% of Waters’ AccQ·Tag Ultra Eluent A. The mobile phase B used Waters’ AccQ·Tag Ultra Eluent B and 10% of water [90:10] (*v*/*v*). The mobile phase C used ultrapure water, and the mobile phase D used Waters’ AccQ·Tag Ultra Eluent B. The flow rate was 700 µL/min, and the detection wavelength was 260 nm. The standards of the amino acids were obtained from Waters (USA). The eluted compounds were identified based on their retention times, their absorption spectra with the authentic standards, and their known, published spectra. The specific contents of the individual amino acids were quantified based on the standard curves and were expressed as a percentage content of DW per sample.

#### 4.5.3. Determination of Catechins

The catechins were determined using HPLC (Waters, Milford, MA, USA), following the national standard GB/T 8313–2018 with minor modifications [48], and the details were as follows. The total catechins were extracted by adding 10 mL of preheated 70% (*v*/*v*) methanol, which had previously been warmed to 70 °C, to 0.1 g of dried pulverized samples. The extraction was carried out in a 70 °C water bath for 15 min. The supernatant was collected through centrifuging at 4000 rpm for 10 min and then filtered through a hydrophilic nylon membrane filter with a 0.22 μm pore size. Afterward, the solution was mixed in a ratio of 1:1 (*v*:*v*) with the stabilizing solution (which included EDTA-2Na and ascorbic acid) and stored at 4 °C for testing. The stabilizing solution was dispensed according to the GB/T 8313–2018.

### 4.6. Statistical Analysis

All the experimental data were presented as the mean with a standard error of the mean (SEM). The data were analyzed with Student’s *t*-test using the IBM SPSS Statistics 27 software (SPSS Inc., Chicago, IL, USA) to estimate the significance; the asterisks represent the significant differences (* *p* < 0.05, ** *p* < 0.01, and *** *p* < 0.001). The diagrams were designed using GraphPad Prism 8 (San Diego, CA, USA). The expression levels in the heat maps are based on the FPKM values that were represented as a normalized Z-score.

## 5. Conclusions

In this study, transcriptome and biochemical analyses were performed to identify the major molecular mechanisms related to leaf color and biochemical component variation in the bud sport HY. The results showed that the reduction in chlorophylls which might result from the intensified expression of genes (SGR, CLH, etc.) related to chlorophyll degradation may contribute to the leaf-color mutation of HY. The decrease in catechins is primarily attributed to the down-regulated expression of key enzymes (CHS, DFR, and ANS) involved in the flavonoid metabolism pathway. These results provide insights into the mechanism of leaf-color mutation and the different contents of quality-related biochemical components; from these insights the theoretical underpinning for the utilization and selective breeding of specific tea cultivars with characterized leaf color may be developed.

## Figures and Tables

**Figure 1 ijms-24-15070-f001:**
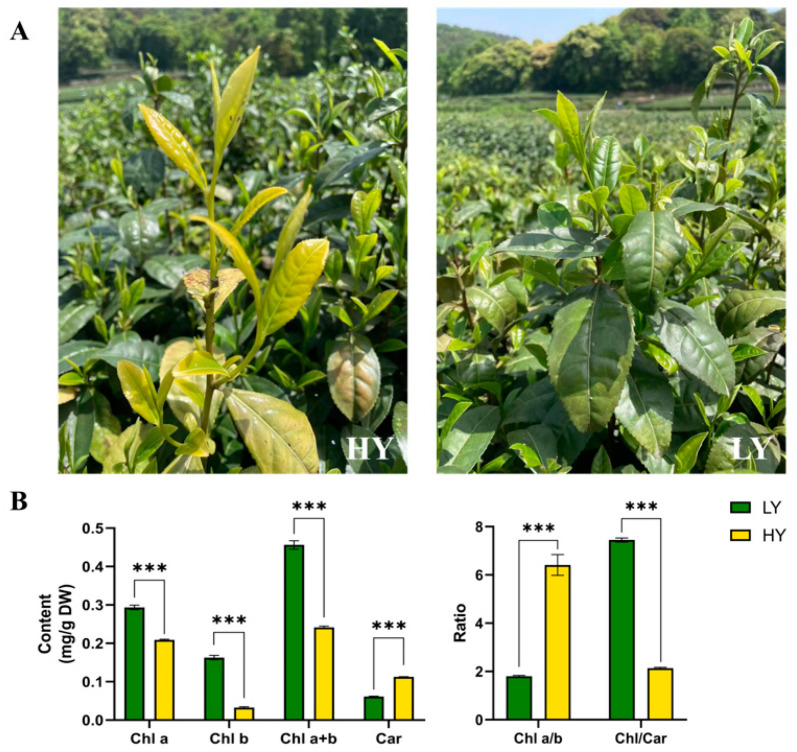
Plant phenotypes (**A**, captured on 16 April 2023) and comparisons in the contents of pigments (**B**) of the HY (the yellow bud mutant shoot) and the LY (the green tea shoot) tea shoots. The plots are presented as mean ± SEM (standard error of the mean, *n* = 3). The significant differences (*** *p* < 0.001) between LY and HY are determined by Student’s *t*-test. Chl a, chlorophyll a; Chl b, chlorophyll b; Chl a+b, the content of total chlorophyll a and chlorophyll b; Car, total carotenoids; Chl a/b, the ratio of Chl a to Chl b; and Chl/Car, the ratio of Chl a+b to Car.

**Figure 2 ijms-24-15070-f002:**
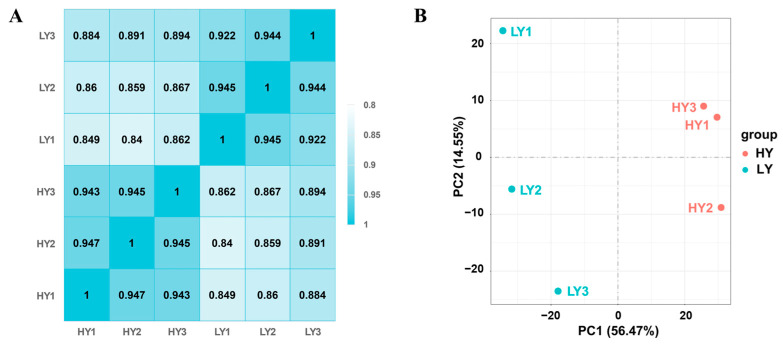
Evaluation of the transcriptome quality in sample reproducibility and inter-group difference. (**A**) Heat map showing Pearson’s correlation coefficients of gene expression among different samples pairwise. (**B**) Principal component analysis (PCA) of the sequencing samples based on gene expression levels using their FPKM values. All the analyses were based on three biological replicates within groups.

**Figure 3 ijms-24-15070-f003:**
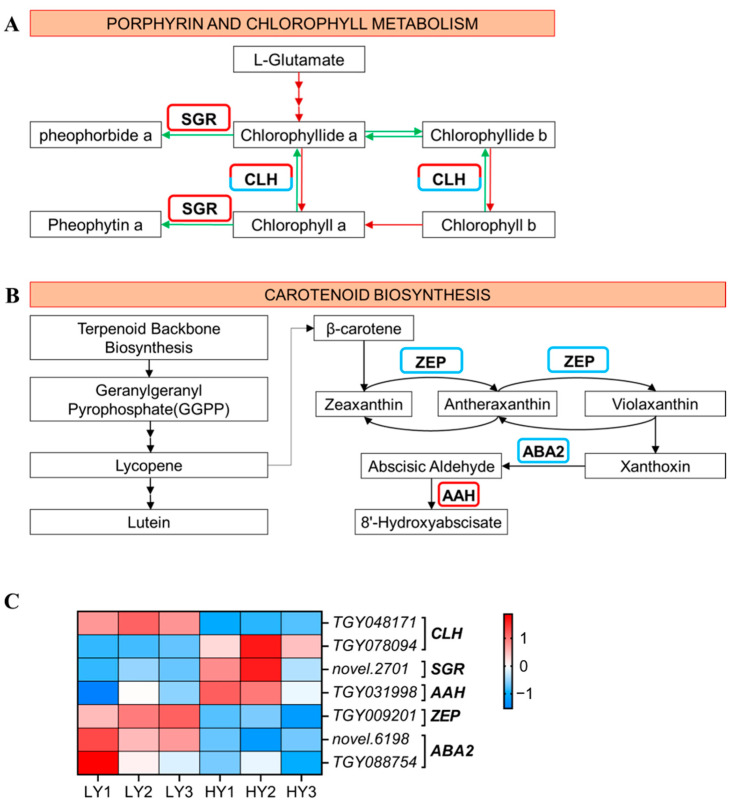
Expression profiles of the DEGs mapped to the porphyrin and chlorophyll metabolism pathway and to the carotenoid biosynthesis pathway. (**A**) Porphyrin and chlorophyll metabolism pathway. The red arrows indicate the pathway of chlorophyll synthesis and green represents chlorophyll decomposition. (**B**) The carotenoid biosynthesis pathway. The up-regulated genes that had higher gene expression levels in HY and lower in LY are marked by a red border, and the down-regulated genes that had lower gene expression levels in HY and higher in LY are blue. (**C**) The cluster heat map of the DEGs in the porphyrin and chlorophyll metabolism and in the carotenoid biosynthesis. The expression levels are based on the FPKM values that were represented as a normalized Z-score. CLH, chlorophyllase; SGR, STAY-GREEN; AAH, (+)-Abscisic acid 8’-hydroxylase; ZEP, zeaxanthin epoxidase; ABA2, xanthoxin dehydrogenase.

**Figure 4 ijms-24-15070-f004:**
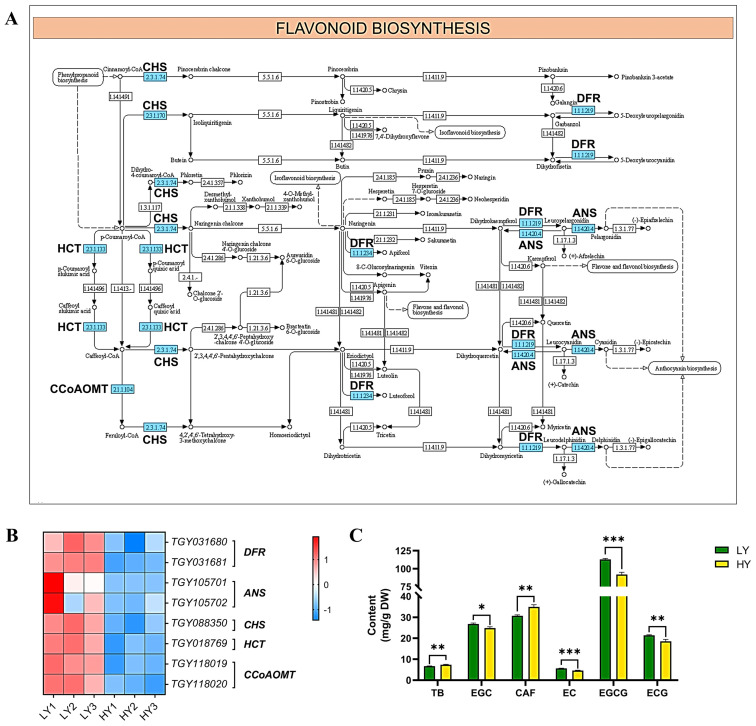
Expression profiles of the DEGs mapped to the flavonoid biosynthesis pathway. (**A**) The flavonoid biosynthesis pathway. In (**A**), the schematic diagram of the flavonoid biosynthesis pathway was downloaded from the KEGG database, and the down-regulated genes or enzymes are marked using a blue rectangle. (**B**) The cluster heat map of the DEGs in flavonoid biosynthesis. The expression levels are based on the FPKM values that were represented as a normalized Z-score. DFR, dihydroflavonol 4-reductase; ANS, anthocyanidin synthase; CHS, chalcone synthase; HCT, shikimate O-hydroxycinnamoyltransferase; and CCoAOMT, caffeoyl-CoA O-methyltransferase. (**C**) Comparisons in the contents of flavonoids. The plots are presented as mean with SEM (*n* = 3). The significant differences (* *p* < 0.05, ** *p* < 0.01, and *** *p* < 0.001) between LY and HY are determined using Student’s *t*-test. * *p* < 0.05 indicates a significant difference (*p* < 0.05) between LY and HY. TB, theobromine; CAF, caffeine; EC, (−)-epicatechin; ECG, (−)-epicatechin gallate; EGC, (−)-epigallocatechin; and EGCG, (−)-epigallocatechin gallate.

**Figure 5 ijms-24-15070-f005:**
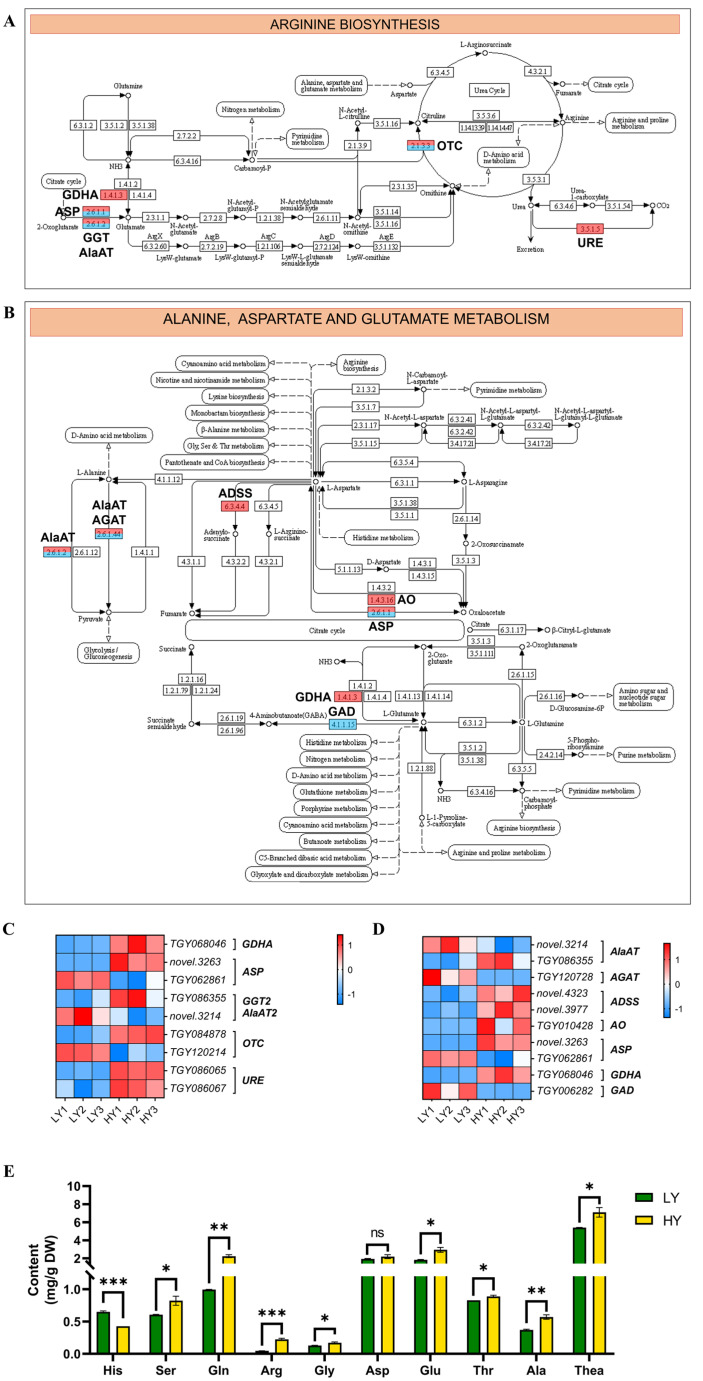
Expression profiles of the DEGs mapped to the amino acids’ metabolism pathway. (**A**) The arginine biosynthesis pathway. (**B**) The alanine, aspartate, and glutamate metabolism pathway. In (**A**,**B**), the schematic diagram of the arginine biosynthesis pathway, and the alanine, aspartate, and glutamate metabolism pathway were downloaded from the KEGG database. The up-regulated genes are marked using a red rectangle and the down-regulated genes are blue. (**C**,**D**) The cluster heat map of the DEGs in the amino acids’ metabolism. The expression levels are based on the FPKM values that were represented as a normalized Z-score. (**E**) Comparisons in the contents of ten free amino acids between HY and LY. The plots are presented as mean with SEM (*n* = 3). The asterisks represent significant differences between LY and HY in amino acid content (ns, not significant; * *p* < 0.05; ** *p* < 0.01; and *** *p* < 0.001). * *p* < 0.05 indicates a significant difference (*p* < 0.05) between LY and HY. GDHA, glutamate dehydrogenase A; ASP, aspartate aminotransferase; GGT2, glutamate--glyoxylate aminotransferase 2; AlaAT2, alanine aminotransferase 2; OTC, ornithine carbamoyltransferase; URE, urease; AGAT, Alanine–glyoxylate aminotransferase; ADSS, adenylosuccinate synthetase; AO, L-aspartate oxidase; His, histidine; Ser, serine; Gln, glutamine; Arg, arginine; Gly, glycine; Asp, aspartic acid; Glu, glutamic acid; Thr, threonine; Ala, alanine; and Thea, theanine.

**Figure 6 ijms-24-15070-f006:**
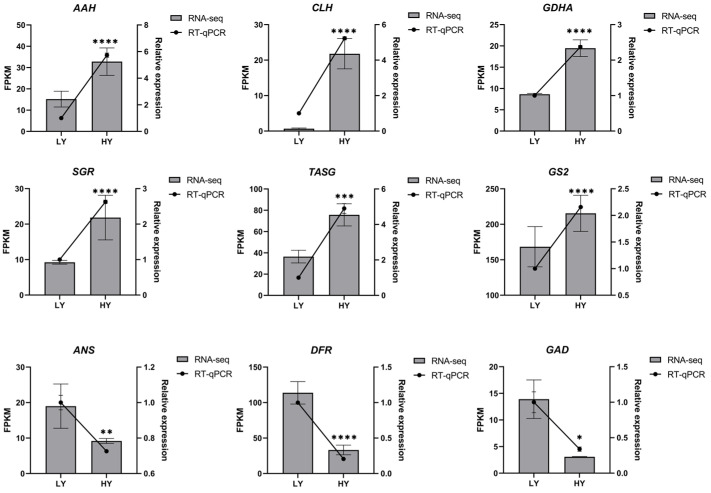
RT-qPCR validation of the nine candidate DEGs. The asterisks represent significant differences between LY and HY in the level of relative expression (* *p* < 0.05, ** *p* < 0.01, *** *p* < 0.001, and **** *p* < 0.0001). * *p* < 0.05 indicates a significant difference (*p* < 0.05) between LY and HY. AAH, (+)-Abscisic acid 8’-hydroxylase; CLH, Chlorophyllase; GDHA, Glutamate dehydrogenase A; SGR, STAY-GREEN; TASG, F-type H+-transporting ATPase subunit gamma; GS2, Glutamine synthetase; ANS, Anthocyanidin synthase; DFR, Dihydroflavonol 4-reductase 2; and GAD, Glutamate decarboxylase.

## Data Availability

Data will be made available on request.

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
