# Peer review of "Transcriptome and Biochemical Analyses of a Chlorophyll-Deficient Bud Mutant of Tea Plant (Camellia sinensis)"

_ijms, 2023, doi:10.3390/ijms242015070_

Round 1

Reviewer 1 Report

In the biological replicates of LY, (Specifically In Figure 2B (PCA), and all the heatmaps) there is a huge variation. Please provide details why the biological replicates of the same sample have this much variation?   

Provide links or citations to the software/databases you used to analyze the transcriptome data.

Please rectify typos and grammatical mistakes.

Line 140-141: rewrite the “there were three DEGs were identified” as “Three DEGs were identified”.

Please carefully revise the paper with an Native English expert.   

Reviewer 2 Report

In my opinion, the study of color mutations in tea from a biochemical and molecular perspective is interesting. In general I think that the work is well developed, although here I indicate in general terms what I consider most relevant and that the information should be included or justified:

- The material and methods section should be completed with all the agronomic details, indicating sampling details, number of plants, samples per plant, etc. At a statistical level, the results shown should be completed by including statistical detail of the significant differences in Figures 5E and 6, as well as their detail in legends and section 4.6 Statistical analysis.

- I think that the results obtained should be confirmed with the new cultivar developed from this mutation 'Zhongcha 133' (accession number: CNA20140553.1). Although it is indicated that the color remains stable, no results are indicated to confirm this statement. The fact that the color remains stable, it can be assumed that it really is a mutation and the color change is not due to nutritional deficiencies or physiopathies in the crop. This is why I consider it relevant that it be justified, including agronomic details and results confirming that the results observed at a biochemical and molecular level are stable in the new cultivar, without nutritional or other deficiencies, including the corresponding controls.

- The importance of this mutation should be highlighted, either commercially or of interest due to its improved (or not) health properties.

- Regarding the introduction and discussion, I think it could be enriched with more background if possible.

- I think the English should be revised, especially in some parts of the manuscript.

I include additional comments in the attached document.

In my opinion, English should be reviewed by a native speaker.
